# Unifying Latent Uncertainty Signals in Large Language Models for Improved Factual Precision

## Abstract

Large Language Models (LLMs) have emerged as powerful tools for knowledge-intensive tasks, yet their tendency to generate factually incorrect or misleading outputs—commonly referred to as hallucinations—poses a fundamental challenge to their reliability. While uncertainty estimation is critical for mitigating such errors, LLMs are not explicitly trained to represent or express uncertainty. In this work, we investigate whether and how uncertainty is implicitly encoded within pretrained models. Through a probing-based analysis, we demonstrate that LLMs internalize multiple distinct and dataset-specific uncertainty signals, which can be extracted as linear directions in their latent space. These signals are most pronounced in intermediate layers, exhibit limited cross-task generalization, and are substantially enhanced by instruction tuning and [IDK]-token training. Building on these findings, we propose Linear Uncertainty Alignment, a novel framework that leverages a unified uncertainty direction to train LLMs to classify their own correctness. Our experiments show that this significantly improves factual precision and reduces hallucination rates under zero-shot evaluation. Together, these results provide new insights into the internal structure of uncertainty in LLMs and introduce a practical method for aligning models toward more trustworthy behavior.

## 1 Introduction

Large Language Models (LLMs) are trained on vast corpora of text data (Brown et al., 2020; Raffel et al., 2020; Chowdhery et al., 2023; Touvron et al., 2023; Le Scao et al., 2023; Jiang et al., 2023a), enabling them to comprehend and generate human language. These training datasets encompass a wide range of written human knowledge, including books, news articles, Wikipedia, and scientific publications. Through this extensive pretraining, LLMs retain significant portions of the information they are exposed to, effectively embedding real-world knowledge within their parameters such that they are able to serve as knowledge repositories (Petroni et al., 2019; Roberts et al., 2020; Cohen et al., 2023a; Pan et al., 2023). This capability allows LLMs to be leveraged in tasks that depend on such knowledge, such as closed-book question answering (Brown et al., 2020; Roberts et al., 2020) and information retrieval (Tay et al., 2022).

Despite their widespread adoption, LLMs are widely known to suffer from 'hallucinations'—a predisposition towards producing outputs that are false or misleading—which significantly undermines their accuracy and trustworthiness (Ji et al., 2023; Manduchi et al., 2024). Hallucinations may manifest in various forms, including factually incorrect statements (Maynez et al., 2020; Devaraj et al., 2022; Tam et al., 2023), internal inconsistencies (Elazar et al., 2021; Mündler et al., 2023), contradictions (Cohen et al., 2024a), or statements lacking clear sources or attribution (Bohnet et al., 2022; Rashkin et al., 2023; Yue et al., 2023).

Uncertainty, however, is a concept that LLMs are not generally known to capture (Yin et al., 2023; Kapoor et al., 2024). At the very least, they are generally not explicitly trained on it. This lack of competency regarding uncertainty, however, often results in misinformation generation, which can be harmful and misleading (Maynez et al., 2020; Devaraj et al., 2022; Tam et al., 2023), as LLMs have a hard time expressing a lack of knowledge both verbally and through their output distribution.

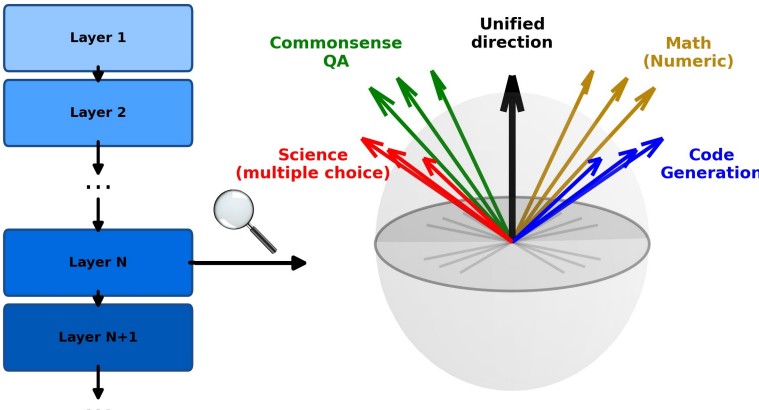

Figure 1: Multiple data-specific linear uncertainty vectors identified at the end of each transformer layer. Vectors are near-orthogonal across topics and align within a topic; a unified vector correlates positively with all.

Some more advanced methods such as instruction tuning (Ouyang et al., 2022; Zhang et al., 2023) during post-training and [IDK] tuning (Cohen et al., 2024b) during pretraining aim, inter alia, to align LLMs to more efficiently express their uncertainty and refrain from misinformation generation. While instruction tuning more generally aligns LLMs with human intent by fine-tuning them on task-specific instructions and corresponding outputs, the model is often also encouraged to refrain from answering questions when the specific answer is not known to it.

In this work, we first propose an analysis mechanism to study the uncertainty captured by large language models (LLMs). Using this mechanism, we show that LLMs internalize a notion of uncertainty during pretraining, which can be extracted via linear probes from their latent representations. Specifically, we identify linear uncertainty vectors—directions in hidden space—that correlate with generation correctness across multiple models and datasets, even without further weight training. This suggests that uncertainty is a learnable, linearly separable concept within LLMs. Our analysis further shows that LLMs do not encode a single unified notion of uncertainty, but rather multiple distinct vectors tied to different datasets or knowledge types. While these vectors are often nearly independent, partial transfer exists—e.g., across mathematics benchmarks. These findings suggest new ways of mitigating hallucination, since inconsistencies between uncertainty signals may underlie unreliable outputs.

Building on these insights, we introduce *Linear Uncertainty Alignment*, a method for aligning models with their internal uncertainty signals by training them to exploit the identified classifiers to predict their own correctness. This alignment substantially improves factual precision in zero-shot evaluation and reinforces that the linear directions correspond to meaningful representations of uncertainty.

In summary, our contributions are twofold: (1) we provide a systematic analysis showing that LLMs implicitly encode multiple forms of uncertainty that can be isolated through linear probes (Section 2), and (2) we present a training framework that leverages these signals to improve factual precision and reduce hallucination (Section 3).

## 2 IDENTIFYING UNCERTAINTY PREDICTORS

### 2.1 FRAMEWORK

In this work, we assume that uncertainty is represented within an LLM's latent space across layers. Specifically, let $\mathbf{h}_i(x)$ denote the hidden state at the end of the $i$-th layer, taken at the last vector (created from the last input token), given input prompt $x$. For each hidden state, we search for a linear vector $\mathbf{u}_i$ such that the classifier $C(x, i) = \mathbf{u}_i^\mathsf{T} \mathbf{h}_i(x) + b_i$ predicts the model's subsequent answer correctness with accuracy significantly above chance. Intuitively, this identifies a linear concept encoding the model's uncertainty about its own outputs.

| Model | OpenBookQA | PopQA | Qampari | ROMQA | SVAMP | StrategyQA | TriviaQA | TruthfulQA |
|-------|-----------|-------|---------|-------|-------|-----------|----------|-----------|
| `Llama-3.2-1B` | 0.534 | 0.857 | 0.634 | 0.750 | 0.729 | 0.689 | 0.716 | 0.737 |
| `Llama-3.2-3B` | 0.590 | 0.793 | 0.734 | 0.583 | 0.750 | 0.608 | 0.742 | 0.600 |
| `Llama-3.1-8B` | 0.644 | 0.757 | 0.630 | 0.763 | 0.711 | 0.684 | 0.757 | 0.722 |
| `Llama-3.1-8B-Instruct` | 0.694 | 0.768 | 0.679 | 0.750 | 0.767 | 0.639 | 0.776 | 0.719 |
| `Mistral-7B-v0.1` | 0.597 | 0.747 | 0.727 | 0.750 | 0.687 | 0.643 | 0.760 | 0.673 |
| `IDK-tuned-Mistral-7B-v0.1` | 0.611 | 0.829 | 0.789 | 0.667 | 0.628 | 0.547 | 0.693 | 0.725 |
| `Qwen2.5-7B` | 0.678 | 0.817 | 0.697 | 0.615 | 0.696 | 0.698 | 0.717 | 0.678 |
| `Qwen3-14B` | 0.743 | 0.833 | 0.630 | 0.596 | 0.789 | 0.561 | 0.782 | 0.699 |
| `Qwen3-14B-Instruct` | 0.619 | 0.771 | 0.861 | 0.655 | 0.767 | 0.711 | 0.756 | 0.726 |

Table 1: Correctness prediction accuracy across 8 of the evaluation datasets (see Table 5 for additional results).

**Linear Uncertainty Search**  Let $M$ be a language model and $\mathcal{D} = \{(q_j, a_j)\}_{j=1}^n$ a dataset of questions and answers. To find $\mathbf{u}_i$ for layer $i$, we train a linear classifier to predict whether $M$'s answer to $q_j$ is correct. Using a training split $\mathcal{D}_{\text{TRAIN}} = \{(q_j, a_j)\}_{j=1}^m, m < n$, we consider the model's predictions and define labels as

$$L(q_j) = \begin{cases} 1 & \text{if } M(q_j) = a_j \\ 0 & \text{otherwise.} \end{cases} \tag{1}$$

This yields $\hat{\mathcal{D}}_{\text{TRAIN}} = \{(q_j, L(q_j))\}_{j=1}^m$. A classifier is then trained on the hidden states $\mathbf{h}_i(q_j)$ to predict $L(q_j)$. With a linear classifier, this corresponds to identifying a direction $\mathbf{u}_i(\mathcal{D})$ in the latent space—referred to as the uncertainty direction for dataset $\mathcal{D}$—along with its learned bias $b_i$.

**Uncertainty Vector as a Predictor**  We evaluate the uncertainty predictor $\mathbf{u}_i(\mathcal{D})$ on held-out test sets from $\mathcal{D}$. Given input $x$, $\mathbf{h}_i(x)$ is the hidden state at layer $i$, and the classifier

$$C_{\mathbf{u}_i(\mathcal{D})}(x) = \begin{cases} \text{INCORRECT} & \text{if } [\mathbf{u}_i(\mathcal{D})]^\intercal \mathbf{h}_i(x) + b_i > 0 \\ \text{CORRECT} & \text{otherwise} \end{cases} \tag{2}$$

predicts whether $M$'s next-token choice is correct. Comparing predictions against the ground truth allows us to compute accuracy, precision, recall, and related metrics.

## 2.2 EXPERIMENTAL SETUP FOR ANALYSIS

To evaluate and utilize our uncertainty identification framework, we consider a series of experiments, for which we first introduce the experimental setup.

**Foundation Models.**  In order to reach general conclusions that are not specific to any particular LLM, in this work we study three different families of models: the Llama family of models (Touvron et al., 2023; Dubey et al., 2024), Mistral (Jiang et al., 2023b), and Qwen (Bai et al., 2023; Yang et al., 2024). Specifically, for Llama, we study `Llama-3.2-1B`, `Llama-3.2-3B`, and `Llama-3.1-8B`, for Mistral, we consider `Mistral-7B-v0.1`, and finally for Qwen, we rely on `Qwen2.5-7B` and `Qwen3-14B`.

**Advanced Models.**  For evaluating the effects of different types of training on the linear uncertainty encodings, we exploit three particular additional models in our experiments. To capture the effect of instruction tuning (Ouyang et al., 2022; Zhang et al., 2023), we use `Llama-3.1-8B-Instruct` and `Qwen3-14B-Instruct`, which both were post-trained using instruction tuning. Furthermore, we follow Cohen et al. (2024b) and use the `IDK-tuned-Mistral-7B-v0.1` model in our experiments to evaluate the effect of [IDK] tuning—a method that essentially adds a new uncertainty token to the model's vocabulary and teaches the model to use it during pretraining by adapting its loss to consider the new token.

**Datasets and Benchmarks.** We utilize 16 QA datasets and benchmarks in both our linear uncertainty search (Section 2.1) and the induced classifier evaluation (Section 2.1). We group them into six thematic categories:

- **Commonsense QA**: *CommonsenseQA* (Talmor et al., 2019), *StrategyQA* (Geva et al., 2021a). These include questions that assess the model's ability to apply everyday reasoning and background knowledge to answer questions beyond surface-level facts.

- **Fact-Lookup and Adversarial QA**: *GranolaEntityQuestions* (Yona et al., 2024), *Natural Questions* (Kwiatkowski et al., 2019), *PopQA* (Mallen et al., 2022), *TriviaQA* (Joshi et al., 2017), *TruthfulQA* (Lin et al., 2021). These consist of questions that test the model's factual recall and resilience to misleading or adversarial question phrasing.

- **List-Output QA**: *QAMPARI* (Amouyal et al., 2023), *RoMQA* (Zhong et al., 2022). Both evaluate whether models can produce comprehensive sets of correct answers, challenging their ability to recall multiple relevant facts simultaneously

- **Science QA (K–12)**: *ARC-Easy* (Clark et al., 2018), *OpenBookQA* (Mihaylov et al., 2018). These focus on elementary school and high-school level science, requiring models to combine factual knowledge with basic reasoning.

- **Math Word Problems**: *GSM8K* (Cobbe et al., 2021), *ASDiv-A* (Miao et al., 2020), *SVAMP* (Patel et al., 2021). These include queries that test models on arithmetic and algebraic reasoning through natural language mathematical problems.

- **Code Generation**: *HumanEval-X* (Zheng et al., 2023), *MBPP* (Austin et al., 2021). We use these datasets to evaluate the ability of models to generate correct and functional software code given natural language programming prompts.

Notably, for each of these, we create a fixed training split, which will be used to derive our uncertainty vectors, and a test split, which will be used to evaluate their performance.

**Linear Uncertainty Search Details.** For every model $M$, transformer layer $i$, and evaluation dataset $\mathcal{D}$, we fit a logistic regression probe on the hidden states $\mathbf{h}_i(x)$ and obtain a single weight vector,

$$\mathbf{u}_i(\mathcal{D}),$$

which serves as the *linear uncertainty direction* for that (layer, dataset) pair.

To obtain a dataset-agnostic baseline, we also train an additional probe on the **concatenation of *all* datasets**. The resulting vector is denoted as

$$\mathbf{u}_i(\mathcal{D}_{\text{UNIFIED}}).$$

**Evaluation.** We evaluate the ability of our identified uncertainty linear vectors to predict the correctness of the model's generation. For this, we consider the following metrics: (i) **Accuracy**: the ratio of correct predictions by the classifier that is induced by the uncertainty linear vector, (ii) **Precision**: the ratio of actually wrong completions by the model among those that the induced classifier predicted to be wrong.

## 2.3 ANALYSIS RESULTS

**A Linear Representation of Uncertainty is Learned during Pretraining.** Tables 1 and 5 report the performance of correctness classifiers, derived from linear uncertainty vectors, across models and datasets (best-performing layers only; full layer-wise analysis in a later section). Despite keeping model weights frozen, we identify linear directions in latent space that yield meaningful correctness predictions, with accuracy well above the 0.5 random baseline. The strongest signals consistently emerge from *upper-intermediate* layers (roughly 2/3 depth), while performance declines in final layers (Figures 8 and 9). This concentration suggests that uncertainty becomes most explicit once knowledge has been consolidated but before it is fully transformed into generation-specific representations. These findings provide strong evidence that uncertainty is encoded in a linearly separable form, concentrated in the middle of the network.

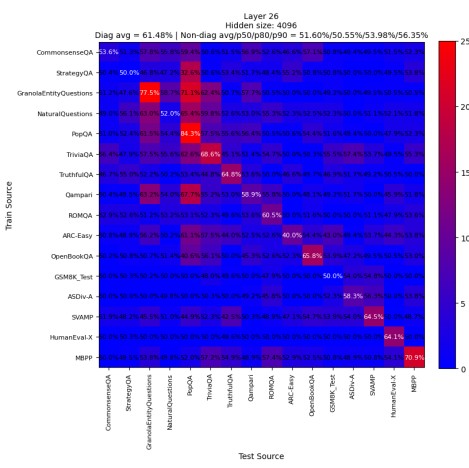

Figure 2: Correctness prediction accuracy results of the classifier induced by $u_{26}(\mathcal{D})$, for datasets $\mathcal{D}$ given on the $y$-axis, using `Llama-3.1-8B`, while testing on the test set for datasets given on the $x$-axis.

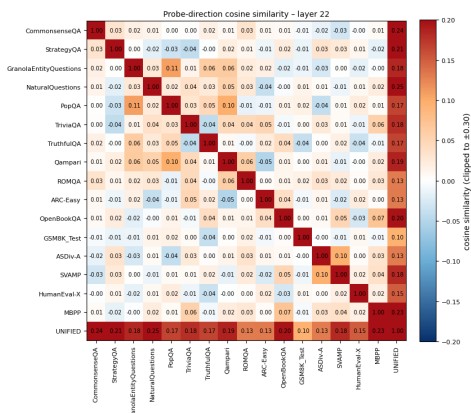

Figure 3: Cosine similarity results across all linear uncertainty vectors at layer number 22 of `Llama-3.1-8B`

**LLMs Learn Multiple Different Linear Uncertainty Vectors.** Linear uncertainty vectors appear across layers but are typically dataset-specific. A classifier trained on $\mathcal{D}_1$ often fails on a different $\mathcal{D}_2$ ($\mathcal{D}_1 \neq \mathcal{D}_2$), and cosine similarity between vectors is often near-zero (Figure 3). Nonetheless, exceptions exist: in domains like **Math Word Problems**, classifiers transfer effectively across datasets such as GSM8K, ASDiv, and SVAMP (Figure 7), sometimes matching or exceeding in-domain accuracy. Thus, while uncertainty is largely dataset-specific, certain domains exhibit shared structures that support generalization.

**A Unified Uncertainty Axis with Universal Positive Alignment.** The *unified* probe $\mathbf{u}_i(\mathcal{D}_{\text{UNIFIED}})$, trained on all datasets, aligns positively with each dataset-specific vector, despite the near-orthogonality of those vectors. One would ordinarily anticipate a mix of positive and negative associations, so this universal positivity reveals a shared component across tasks. Conceptually, this indicates a generalized axis of uncertainty (Figure 1), offering both theoretical insight and a methodological tool for surfacing common uncertainty structure.

**Fine-Tuning, Not Scale, Enhances Uncertainty Representation.** Scaling does little to improve uncertainty prediction: larger Llama models perform only marginally better than smaller ones (Figure 9). In contrast, specialized training substantially boosts performance. Instruction tuning improves the accuracy and shifts peak performance to earlier layers, while [IDK] tuning not only increases accuracy but also reduces overconfidence, yielding more reliable predictors. Both enhance cross-dataset transfer, underscoring that fine-tuning—not scale—is key to strengthening uncertainty representations. These results suggest that tuning strategies can explicitly guide models to align latent uncertainty signals with observable correctness, whereas sheer model size does not.

## 3 LINEAR UNCERTAINTY ALIGNMENT

As a consequence of our previous analysis (Section 2), we propose a novel tuning mechanism designed to explicitly align language models with their own internal uncertainty representations. The key idea is to leverage the dataset-specific uncertainty vectors $\mathbf{u}_i(\mathcal{D})$ identified in Section 2.1 and integrate them into the training objective. By doing so, the model is encouraged not only to predict correct answers but also to recognize and classify the correctness of its own generations.

## 3.1 DUAL-OBJECTIVE TRAINING

Our method tunes the model with two complementary objectives:

**Standard Cross-Entropy Loss.** As in conventional fine-tuning, the model is trained to maximize the likelihood of gold answers. Given a dataset $\mathcal{D} = \{(q_j, a_j)\}_{j=1}^n$, the cross-entropy loss is defined as

$$\mathcal{L}_{\text{CE}} = -\frac{1}{n} \sum_{j=1}^n \log P_\theta(a_j \mid q_j), \tag{3}$$

where $P_\theta(a_j \mid q_j)$ denotes the probability assigned by the model with parameters $\theta$ to the correct answer $a_j$ given question $q_j$.

**Uncertainty Classification Loss.** To align the model with its own uncertainty, we introduce a secondary objective that encourages hidden states to reflect correctness predictions. For each input $q_j$, let $\mathbf{h}_i(q_j)$ denote the hidden state of layer $i$, and let $L(q_j) \in \{0, 1\}$ be the correctness label defined in Equation 1. Using the fixed uncertainty vector $\mathbf{u}_i(\mathcal{D})$ and bias $b_i$, the predicted correctness is given by

$$\hat{L}(q_j) = \sigma\left(\mathbf{u}_i(\mathcal{D})^\mathsf{T} \mathbf{h}_i(q_j) + b_i\right), \tag{4}$$

where $\sigma(\cdot)$ is the logistic sigmoid function. The corresponding classification loss is

$$\mathcal{L}_{\text{U}} = -\frac{1}{n} \sum_{j=1}^n \left[ L(q_j) \log \hat{L}(q_j) + (1 - L(q_j)) \log(1 - \hat{L}(q_j)) \right]. \tag{5}$$

## 3.2 UNIFIED OBJECTIVE

The final training objective combines both components as

$$\mathcal{L} = \mathcal{L}_{\text{CE}} + \lambda \cdot \mathcal{L}_{\text{U}}, \tag{6}$$

where $\lambda$ is a tunable hyperparameter controlling the trade-off between improving factual correctness and aligning with the model's internal uncertainty representation.

This dual-objective formulation encourages the model to (a) produce accurate answers where possible, while (b) learning to reflect and calibrate its uncertainty in line with the fixed, data-derived uncertainty vectors. As we show in Section 4, models tuned with this method exhibit improved factual precision and reduced hallucination rates under zero-shot evaluation.

# 4 EXPERIMENTS AND RESULTS

We proceed by detailing the setup underlying the experimental evaluation of our proposed *Linear Uncertainty Alignment* method, followed by a discussion of the corresponding results and conclusions.

## 4.1 EXPERIMENTAL SETUP

Our experimental design builds directly on the models, datasets, and evaluation metrics introduced in Section 2. Here, we describe the details specific to assessing our alignment method.

**Instruction-Based Evaluation Protocol.** To probe both factual precision and calibrated abstention behavior, we employ an instruction-driven evaluation format. The instruction is designed to explicitly invite models to abstain when uncertain, thereby testing their ability to leverage internal uncertainty representations. Specifically, each test instance is presented in the form:

> "Please answer the following question. Please answer with *I don't know the answer* in cases where you're not certain in your answer. The question is: ..."

| Dataset | Llama-3.2-1B | | | | | | Llama-3.2-3B | | | | | | Llama-3.1-8B | | | | | |
|---------|---|---|---|---|---|---|---|---|---|---|---|---|---|---|---|---|---|---|
| | Tuned | | | +Aligned | | | Tuned | | | +Aligned | | | Tuned | | | +Aligned | | |
| | P | R | F1 | P | R | F1 | P | R | F1 | P | R | F1 | P | R | F1 | P | R | F1 |
| CommonsenseQA | 40.5 | **25.0** | 30.9 | **50.7** | 23.9 | **32.5** | 38.9 | **28.7** | 33.0 | **50.5** | 28.1 | **36.2** | 45.3 | **34.0** | 38.9 | **57.2** | 32.7 | **41.6** |
| Natural Questions | 27.0 | **18.9** | 22.2 | **36.1** | 18.1 | **24.1** | 28.5 | **21.7** | 24.7 | **36.9** | 20.4 | **26.0** | 31.7 | **25.5** | 28.3 | **40.4** | 24.0 | **30.1** |
| TriviaQA | 56.2 | **47.0** | 51.2 | **65.9** | 46.2 | **54.4** | 58.5 | **52.7** | 55.5 | **66.7** | 50.9 | **57.8** | 59.8 | **58.9** | 59.3 | **68.1** | 57.8 | **62.5** |
| PopQA | 45.1 | **32.0** | 37.4 | **55.0** | 30.8 | **39.6** | 45.7 | **36.2** | 40.4 | **56.9** | 34.9 | **43.2** | 48.4 | **42.5** | 45.3 | **60.1** | 39.9 | **48.0** |
| TruthfulQA | **35.6** | 27.9 | 31.3 | 44.8 | **26.8** | **33.4** | 35.5 | 31.6 | 33.4 | 45.0 | **30.1** | **36.1** | 39.6 | 35.0 | 37.1 | 49.8 | **34.2** | **40.7** |
| GSM8K | 46.9 | **39.7** | 43.0 | **54.7** | 39.2 | **45.8** | 48.6 | **42.5** | 45.4 | **56.3** | 41.6 | **47.9** | 53.0 | **46.8** | 49.7 | **61.8** | 46.3 | **53.0** |
| Average | 41.9 | **31.7** | 36.0 | **51.2** | 30.8 | **38.3** | 42.6 | **35.6** | 38.7 | **52.0** | 34.3 | **41.2** | 46.3 | **40.5** | 43.1 | **56.2** | 39.2 | **46.0** |

Table 2: Comparison of precision (P), recall (R), and F1-score between tuned and uncertainty-aligned LLaMA models across evaluation datasets. For each dataset and metric, the better result is highlighted in bold.

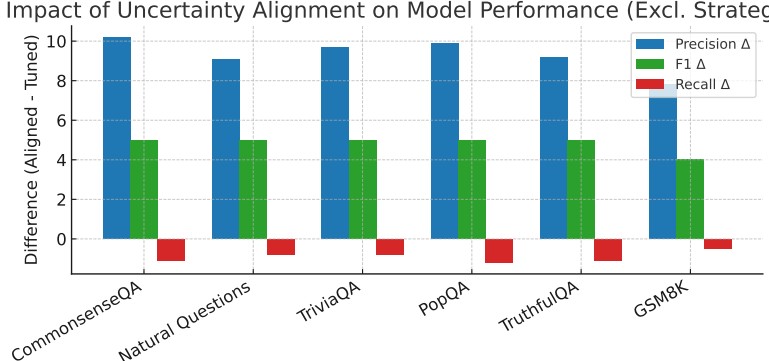

Figure 4: Comparison of precision, recall, and F1-score between tuned and uncertainty-aligned LLaMA models (`Llama-3.2-1B`, `Llama-3.2-3B`, `Llama-3.1-8B`). Alignment consistently improves **precision** and **F1**, with only a minor decrease in recall.

**Tuning Variants.** For each foundation model considered, we fine-tune the model under the dual-objective formulation from Section 3. The tuning is conducted separately with respect to:

1. each layer-specific uncertainty vector $\mathbf{u}_i(\mathcal{D})$ identified in Section 2;
2. the unified uncertainty direction $\mathbf{u}_{\text{uni}}$ obtained by aggregating datasets (Section 2.3).

These two variants allow us to directly compare the impact of alignment to individual vectors versus the global, unified representation.

**Evaluation.** Each tuned model variant is evaluated across all benchmark datasets used in our earlier analysis, following the same correctness, precision, and hallucination metrics. We thus obtain a comprehensive picture of how uncertainty alignment impacts model behavior across diverse tasks. In particular, we measure improvements in factual precision, the reduction of spurious hallucinations, and the calibrated use of abstentions (*I don't know*) as a function of the tuning vector employed.

## 4.2 EVALUATION RESULTS

**Per-Dataset Evaluation.** Our first evaluation examines the effect of the proposed method on factual precision in a dataset-specific setup. Concretely, for each dataset, we train a separate model using its training split and then evaluate performance on the corresponding held-out test split. Importantly, we choose the fixed uncertainty vector to be the one corresponding to the layer for which the corresponding uncertainty vector produces the best classification results out of all the model layers. Across all model sizes and datasets, we observe a consistent pattern: alignment markedly boosts precision, with average gains of nearly 10 points, while recall is only moderately affected (Tables 2 and 6, Figure 4). This balance results in substantial improvements in F1, confirming that the method not only reduces factual errors but also preserves the model's ability to retrieve correct answers. For example, on CommonsenseQA, precision improves from 45.3 to 57.2 for the 8B model, with recall remains at a competitive 32.7, leading to an F1 increase from 38.9 to 41.6. Similar trends are observed across Natural Questions, TriviaQA, PopQA, TruthfulQA, and GSM8K, where precision gains consistently outweigh the relatively small recall drops, yielding stronger overall F1 scores.

| Dataset | LLaMA-1B | | | | | | LLaMA-3B | | | | | | LLaMA-8B | | | | | |
| | Tuned | | | +Aligned | | | Tuned | | | +Aligned | | | Tuned | | | +Aligned | | |
| | P | R | F1 | P | R | F1 | P | R | F1 | P | R | F1 | P | R | F1 | P | R | F1 |
|---|---|---|---|---|---|---|---|---|---|---|---|---|---|---|---|---|---|---|
| CommonsenseQA | 36.1 | **25.7** | 30.0 | **43.8** | 23.5 | **31.0** | 33.4 | **27.0** | 29.9 | **41.5** | 26.8 | **32.6** | 41.0 | **31.3** | 35.5 | **47.6** | 32.5 | **38.6** |
| StrategyQA | 35.0 | **20.3** | 25.7 | 32.2 | **24.1** | **27.7** | 34.6 | 26.5 | 30.0 | **36.0** | 24.9 | 29.5 | 33.9 | **29.0** | 31.3 | **38.0** | 25.1 | 30.3 |
| Natural Questions | 24.6 | **17.1** | 20.2 | **29.6** | 18.7 | 23.0 | 25.4 | **19.9** | 22.3 | **31.0** | 19.8 | **24.4** | 32.0 | **23.2** | 26.9 | **34.4** | 24.8 | **28.8** |
| TriviaQA | 56.0 | **46.1** | 50.6 | **60.7** | 44.9 | **51.6** | 58.3 | **53.8** | 56.0 | **61.1** | 50.3 | **56.0** | 58.3 | **58.9** | 58.6 | **65.0** | 56.1 | **60.4** |
| PopQA | 47.3 | **32.3** | **38.4** | **48.2** | 30.1 | 37.5 | 43.9 | 33.9 | 38.3 | **48.9** | 34.3 | 40.3 | 50.2 | **43.6** | 46.7 | **51.4** | 40.0 | 45.0 |
| TruthfulQA | 33.7 | **28.2** | 30.7 | **38.1** | 26.0 | **31.1** | 37.3 | **32.5** | 34.7 | **38.7** | 29.4 | 33.4 | 39.1 | **33.9** | 36.3 | **42.5** | 32.5 | **36.6** |
| GSM8K | 41.7 | **39.9** | 40.8 | **50.4** | 38.0 | **43.6** | 45.3 | **41.7** | 43.4 | **52.8** | 40.6 | **46.0** | 50.4 | **38.0** | 43.3 | **57.0** | 44.5 | **50.1** |

Table 3: Performance comparison between the baseline tuned LLaMA models and our **Aligned** variants trained with the unified uncertainty vector as the classification signal in the auxiliary loss. Results are reported on all evaluation datasets in terms of precision (P), recall (R), and F1-score. For each dataset and metric, the better score between **Tuned** and **+Aligned** is highlighted in **bold**.

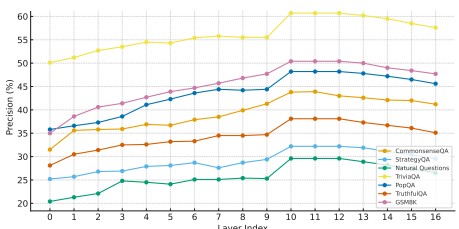

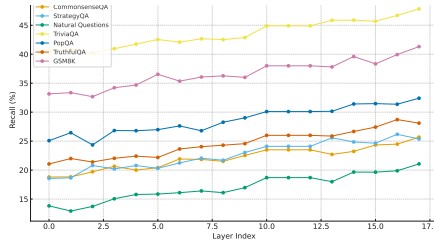

(a) Precision per layer across datasets.       (b) Recall per layer across datasets.

Figure 5: Layer-wise evaluation of `Llama-3.2-1B` models aligned using the different layer-specific uncertainty vectors. Results are shown across all datasets in terms of (a) precision and (b) recall. The curves demonstrate consistent improvements near the alignment layers, indicating that the extracted vectors capture meaningful uncertainty signals.

These findings provide strong empirical support for our central hypothesis: the linear uncertainty direction we identified indeed captures meaningful uncertainty in the model's latent space. By aligning generations along this axis, the model becomes more calibrated with respect to factual correctness, which manifests as higher precision and better overall reliability.

**Unified Vector results.** In contrast to our above per-dataset alignment experiments, where a separate model was tuned for each dataset-specific uncertainty vector, here we trained a single model by leveraging the unified uncertainty vector as the auxiliary classification signal. For both the baseline tuned models and our aligned variants, the training set was constructed by randomly sampling 10% of each dataset's original training split used in the previous experiments, ensuring a fair and consistent comparison. As shown in Tables 3 and 7, the unified alignment consistently improves precision and F1 scores relative to the tuned baseline across nearly all datasets, while recall remains competitive. Importantly, the fact that a single unified vector—learned once and applied across heterogeneous tasks—leads to systematic gains provides strong evidence that this vector indeed captures and represents the general concept of uncertainty within the model's latent space.

### 4.3 FURTHER ANALYSIS

**Layer Analysis.** Figure 5 shows per-layer trends in precision and recall across all datasets. Performance is consistently lowest at the embedding layer (layer 0), increases through intermediate layers, and peaks around layers 10–12. Beyond this range, precision and recall either plateau or decline slightly, suggesting that mid-layer representations capture the most informative uncertainty signal for alignment. These findings are compatible with our earlier analysis results, further reinforcing the conclusion that the unified uncertainty vector captures the underlying concept effectively (see Appendix C).

**Uncertainty Classification Loss Only.** In this ablation study, we trained models using only the auxiliary classification loss based on the uncertainty vector, while omitting the standard cross-entropy objective. We then compared their performance to our full +Aligned method across all datasets. The results show that removing cross-entropy leads to a substantial drop in recall, in some cases exceeding

| Dataset | LLaMA-1B | | | | | | LLaMA-3B | | | | | | LLaMA-8B | | | | | |
|---|---|---|---|---|---|---|---|---|---|---|---|---|---|---|---|---|---|---|
| | +Aligned | | | Ablation | | | +Aligned | | | Ablation | | | +Aligned | | | Ablation | | |
| | P | R | F1 | P | R | F1 | P | R | F1 | P | R | F1 | P | R | F1 | P | R | F1 |
| CommonsenseQA | 43.8 | **23.5** | **31.0** | **44.2** | 17.9 | 25.3 | 41.5 | **26.8** | **32.6** | 41.2 | 19.7 | 26.5 | **47.6** | **32.5** | **38.6** | 47.0 | 24.8 | 32.3 |
| StrategyQA | 32.2 | **24.1** | **27.7** | **33.0** | 16.0 | 21.2 | **36.0** | **24.9** | **29.5** | 35.6 | 17.8 | 23.2 | **38.0** | **25.1** | **30.3** | 37.7 | 19.4 | 25.2 |
| Natural Questions | 29.6 | **18.7** | **23.0** | **30.1** | 13.5 | 18.6 | **31.0** | **19.8** | **24.4** | 30.6 | 14.2 | 19.6 | **34.4** | **24.8** | **28.8** | 33.9 | 17.1 | 22.4 |
| TriviaQA | 60.7 | **44.9** | **51.6** | **61.2** | 32.5 | 40.6 | 61.1 | **50.3** | **56.0** | 60.9 | 36.4 | 45.2 | **65.0** | **56.1** | **60.4** | 64.7 | 41.2 | 49.9 |
| PopQA | **48.2** | **30.1** | **37.5** | 47.8 | 19.7 | 27.9 | 48.9 | **34.3** | **40.3** | **49.2** | 21.8 | 29.8 | **51.4** | **40.0** | **45.0** | 51.0 | 28.4 | 36.6 |
| TruthfulQA | 38.1 | **26.0** | **31.1** | **38.5** | 17.9 | 24.2 | 38.7 | **29.4** | **33.4** | **39.0** | 21.0 | 27.1 | **42.5** | **32.5** | **36.6** | 42.2 | 23.7 | 29.9 |
| GSM8K | 50.4 | **38.0** | **43.6** | **50.7** | 25.7 | 32.6 | **52.8** | **40.6** | **46.0** | 52.3 | 27.1 | 35.4 | **57.0** | **44.5** | **50.1** | 56.6 | 29.4 | 38.3 |

Table 4: Ablation study: comparison between our full **+Aligned** method (with cross-entropy and classification loss) and the **Ablation** variant trained only with the classification loss. Results are reported on all evaluation datasets in terms of precision (P), recall (R), and F1-score. The better score for each dataset and metric is highlighted in **bold**.

25%, while precision is occasionally comparable or slightly higher. However, the overall F1 scores consistently favor the full method, highlighting that the cross-entropy loss is crucial for maintaining balanced predictions and preventing the model from becoming overly conservative.

## 5 RELATED WORK

**Model Calibration.** Our analysis is closely related to the challenge of model calibration (Guo et al., 2017): providing a measure of the probability that a prediction is incorrect alongside the prediction itself. Factual error detection can be viewed as a variation of calibration, where instead of a continuous probability, we output a binary judgment of correctness. Common approaches include transformations of model logits (Desai & Durrett, 2020; Jiang et al., 2021) and uncertainty-based methods (e.g., see Kuhn et al., 2023). Recent work explores supervised calibration with LMs, using fine-tuning (Kadavath et al., 2022; Lin et al., 2022), in-context learning (Cohen et al., 2023a; Alivanistos et al., 2022), zero-shot instruction methods (Cohen et al., 2023b), and consistency sampling (Yoran et al., 2023). Other studies leverage internal states for certainty classification (Azaria & Mitchell, 2023), introduce special tokens for unanswerable inputs (Lu et al., 2022), or design datasets for refusal tuning (Zhang et al., 2024). Our work instead analyzes the dynamics of uncertainty encoding in pretrained and calibrated models.

**Mechanistic Interpretability** Recent work has been aiming to identify circuits and features within models that correspond to interpretable concepts such as factual recall, syntax, or positional reasoning (Olsson et al., 2022; Yu et al., 2023). For instance, tools such as SAE (Sparse Autoencoders) have been used to isolate human-interpretable features from residual stream activations (Meng et al., 2022). Other studies explore how knowledge is stored and manipulated across layers, such as tracing factual associations or memorized content to specific directions in the latent space (Geva et al., 2021b; Gurnee et al., 2023; Geva et al., 2023; Yu et al., 2024). Despite promising progress, full mechanistic understanding remains an open challenge due to the scale and complexity of modern models.

## 6 CONCLUSION

Our systematic analysis of uncertainty representation in Large Language Models reveals four key findings: (1) linear uncertainty representations are learned during pretraining, with probing classifiers achieving above-chance accuracy in predicting answer correctness, most pronounced in upper-intermediate layers around two-thirds of model depth; (2) models acquire multiple, dataset-specific uncertainty vectors that are largely orthogonal yet show meaningful cross-dataset generalization within task families; (3) despite this near-orthogonality, a unified uncertainty axis aligns positively with all dataset-specific directions, revealing a shared component across tasks; and (4) specialized training strategies—instruction tuning and [IDK] tuning—substantially enhance uncertainty representation, while scaling alone provides only marginal gains.

Building on the third finding, we introduce Linear Uncertainty Alignment, a training method that teaches models to classify their own correctness through dual objectives. This approach improves factual precision while maintaining competitive recall, showing that uncertainty is not only learnable and accessible but also deliberately alignable for building more trustworthy language models.

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

## A  LIMITATIONS

While our analysis provides compelling evidence for the existence of linearly accessible uncertainty representations in LLMs, it is limited to linear probes and does not explore more complex, nonlinear structures that may further explain model behavior. Our evaluation focuses on a fixed set of models and datasets, which, although diverse, may not capture the full variability seen in real-world applications or domain-specific tasks. Additionally, correctness is treated as a proxy for uncertainty, which may not fully align with how uncertainty manifests in open-ended or ambiguous generation scenarios. Finally, the performance of our classifiers may also be influenced by dataset-specific biases, potentially limiting generalizability.

## B  COMPUTATIONAL RESOURCES

In our experiments we use one NVIDIA A100 80G GPU.

## C  ADDITIONAL RESULTS

Here we provide additional noteworthy results from our experiments.

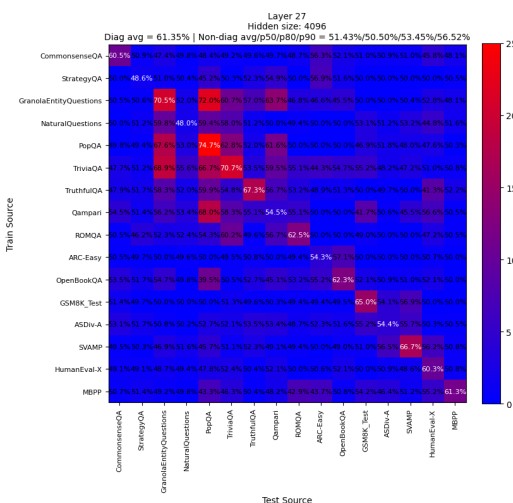

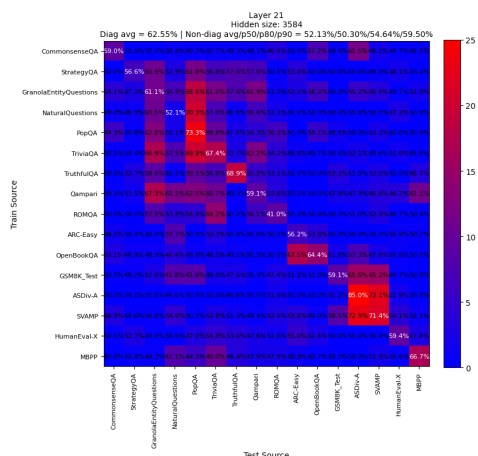

Figure 6: Correctness prediction accuracy results of the classifier induced by $u_{27}(\mathcal{D})$, for datasets $\mathcal{D}$ given on the $y$-axis, using `Mistral-7B-v0.1`, while testing on the test set for datasets given on the $x$-axis.

Figure 7: Correctness prediction accuracy results of the classifier induced by $u_{21}(\mathcal{D})$, for datasets $\mathcal{D}$ given on the $y$-axis, using `Qwen2.5-7B`, while testing on the test set for datasets given on the $x$-axis..

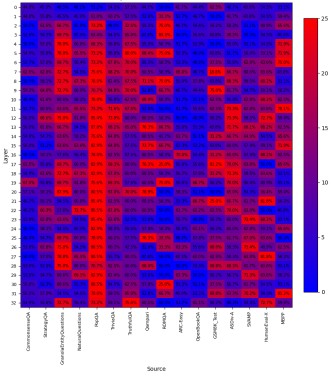

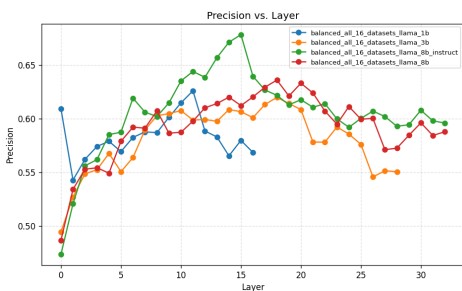

Figure 8: Accuracy results of `Mistral-7B-v0.1` across all model layers and datasets. Here the induced classifiers were tested on the same dataset (but different split) as they were searched on.

Figure 9: Correctness prediction precision averaged over all datasets of the induced classifier, considering the Llama family: `Llama-3.2-1B`, `Llama-3.2-3B`, `Llama-3.1-8B`, and `Llama-3.1-8B-Instruct`.

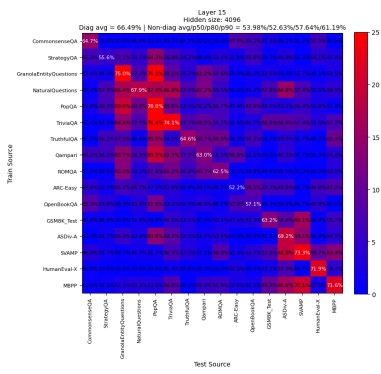

Figure 10: Correctness prediction accuracy results of the classifier induced by $u_{15}(\mathcal{D})$, for datasets $\mathcal{D}$ given on the $y$-axis, using `Llama-3.1-8B-Instruct`, while testing on the test set for datasets given on the $x$-axis.

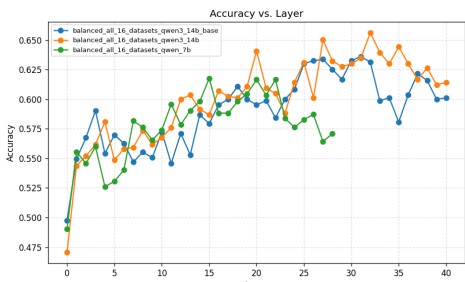

Figure 11: Correctness prediction accuracy averaged over all datasets of the induced classifier, considering the Qwen family: `Qwen2.5-7B`, `Qwen3-14B`, and `Qwen3-14B-Instruct`

| Model | ARC-Easy | ASDiv-A | CommonsenseQA | GSM8K | GranolaEntityQuestions | HumanEval-X | MBPP | NaturalQuestions |
|---|---|---|---|---|---|---|---|---|
| Llama-3.2-1B | 0.535 | 0.670 | 0.625 | 0.444 | 0.789 | 0.708 | 0.769 | 0.600 |
| Llama-3.2-3B | 0.710 | 0.648 | 0.598 | 0.688 | 0.790 | 0.732 | 0.641 | 0.675 |
| Llama-3.1-8B | 0.657 | 0.667 | 0.649 | 0.577 | 0.763 | 0.692 | 0.722 | 0.590 |
| Llama-3.1-8B-Instruct | 0.652 | 0.885 | 0.667 | 0.737 | 0.705 | 0.781 | 0.728 | 0.655 |
| Mistral-7B-v0.1 | 0.657 | 0.691 | 0.709 | 0.550 | 0.782 | 0.707 | 0.707 | 0.630 |
| IDK-tuned-Mistral-7B-v0.1 | 0.600 | 0.750 | 0.571 | 0.688 | 0.758 | 0.545 | 0.688 | 0.673 |
| Qwen2.5-7B | 0.750 | 0.800 | 0.718 | 0.682 | 0.704 | 0.578 | 0.648 | 0.750 |
| Qwen3-14B | 0.727 | 0.786 | 0.655 | 0.878 | 0.738 | 0.800 | 0.694 | 0.651 |
| Qwen3-14B-Instruct | 0.800 | 0.750 | 0.638 | 0.702 | 0.770 | 0.688 | 0.625 | 0.674 |

Table 5: Correctness prediction accuracy across the remaining datasets, complementing the results given in Table 1.

| Dataset | LLaMA-1B | | | | | | LLaMA-3B | | | | | | LLaMA-8B | | | | | |
|---|---|---|---|---|---|---|---|---|---|---|---|---|---|---|---|---|---|---|
| | Tuned | | | +Aligned | | | Tuned | | | +Aligned | | | Tuned | | | +Aligned | | |
| | P | R | F1 | P | R | F1 | P | R | F1 | P | R | F1 | P | R | F1 | P | R | F1 |
| GranolaEntityQ | 33.1 | **22.4** | 26.7 | **34.7** | 21.0 | 26.6 | 32.8 | **25.1** | 28.5 | **35.2** | 24.3 | **28.9** | 36.0 | **28.7** | 32.0 | **38.1** | 27.1 | **31.7** |
| QAMPARI | 29.4 | **19.8** | 23.6 | **31.2** | 18.7 | **23.7** | 28.5 | **22.6** | 25.2 | **32.0** | 21.8 | **26.0** | 33.1 | **27.2** | 29.9 | **35.0** | 26.1 | **29.8** |
| RoMQA | 26.8 | **16.5** | 20.4 | **28.3** | 15.1 | **19.7** | 27.4 | **19.7** | 22.8 | **29.1** | 18.9 | **22.8** | 30.6 | **24.5** | 27.1 | **32.8** | 23.3 | **27.1** |
| ARC-Easy | 42.2 | **31.6** | 36.1 | **44.0** | 30.4 | **36.1** | 45.1 | **37.2** | 40.7 | **47.0** | 35.8 | **40.6** | 48.2 | **41.0** | 44.3 | **49.9** | 39.6 | **44.2** |
| OpenBookQA | 39.0 | **27.3** | 32.1 | **40.5** | 26.0 | **31.6** | 38.7 | **31.8** | 34.9 | **41.2** | 30.7 | **35.2** | 42.5 | **35.4** | 38.6 | **44.3** | 34.0 | **38.4** |
| ASDiv-A | 34.8 | **28.1** | 31.1 | **37.1** | 26.5 | **30.9** | 36.5 | **31.2** | 33.6 | **39.0** | 30.1 | **34.0** | 40.2 | **34.0** | 36.9 | **42.5** | 32.7 | **36.8** |
| SVAMP | 32.7 | **26.4** | 29.2 | **34.5** | 25.0 | **29.1** | 33.9 | **29.3** | 31.4 | **36.1** | 28.1 | **31.9** | 37.5 | **32.1** | 34.6 | **39.8** | 30.5 | **34.6** |
| HumanEval-X | 21.0 | **12.4** | 15.5 | **23.2** | 11.5 | **15.9** | 22.6 | **15.8** | 18.6 | **24.0** | 14.9 | **18.4** | 25.8 | **19.7** | 22.3 | **27.6** | 18.4 | **22.1** |
| MBPP | 26.5 | **18.1** | 21.5 | **27.3** | 17.2 | **21.3** | 27.1 | **20.6** | 23.5 | **28.8** | 19.8 | **23.6** | 29.9 | **24.0** | 26.6 | **31.5** | 22.7 | **26.4** |

Table 6: Performance comparison on the remaining benchmarks not included in Table 3. Consistent with earlier results, the **+Aligned** models typically improve recall and F1 across datasets, while tuned models sometimes retain slightly higher precision.

| Dataset | LLaMA-1B | | | | | | LLaMA-3B | | | | | | LLaMA-8B | | | | | |
|---|---|---|---|---|---|---|---|---|---|---|---|---|---|---|---|---|---|---|
| | Tuned | | | +Aligned | | | Tuned | | | +Aligned | | | Tuned | | | +Aligned | | |
| | P | R | F1 | P | R | F1 | P | R | F1 | P | R | F1 | P | R | F1 | P | R | F1 |
| GranolaEntityQ | 30.5 | **18.6** | 23.1 | **34.8** | 20.1 | 25.5 | 33.0 | **22.0** | 26.4 | **36.7** | 23.5 | **28.7** | 37.9 | **26.0** | 31.0 | **41.0** | 27.1 | **32.7** |
| QAMPARI | 25.2 | **14.7** | 18.6 | **28.5** | 15.9 | 20.4 | 27.1 | **17.5** | 21.3 | **30.2** | 19.0 | **23.2** | 30.5 | **22.0** | 25.6 | **34.1** | 23.5 | **27.9** |
| RoMQA | 21.0 | **13.2** | 16.2 | **24.7** | 14.0 | 18.0 | 22.4 | **15.9** | 18.6 | **26.3** | 16.8 | **20.6** | 27.0 | **18.3** | 21.9 | **30.8** | 19.6 | **23.9** |
| ARC-Easy | 43.2 | **34.5** | 38.4 | **46.0** | 33.1 | 38.7 | 45.0 | **36.8** | 40.5 | **49.1** | 37.2 | **42.3** | 50.3 | **41.0** | 45.2 | **53.0** | 42.1 | **46.9** |
| OpenBookQA | 39.5 | **26.3** | 31.6 | **41.7** | 27.5 | 33.0 | 40.9 | **28.0** | 33.2 | **44.3** | 29.6 | **35.4** | 46.0 | **33.1** | 38.6 | **48.5** | 34.4 | **40.3** |
| ASDiv-A | 37.8 | **29.4** | 33.1 | **42.2** | 30.7 | 35.6 | 39.4 | **31.8** | 35.1 | **44.0** | 33.0 | **37.9** | 45.5 | **36.5** | 40.5 | **49.0** | 37.6 | **42.5** |
| SVAMP | 34.0 | **25.0** | 29.0 | **39.1** | 26.5 | 31.5 | 35.8 | **28.2** | 31.6 | **41.3** | 29.7 | **34.7** | 43.0 | **33.0** | 37.4 | **47.2** | 34.5 | **39.9** |
| HumanEval-X | 20.5 | **12.0** | 15.1 | **23.8** | 13.1 | 17.0 | 21.7 | **13.7** | 16.7 | **25.5** | 14.9 | **19.2** | 26.1 | **16.5** | 20.2 | **29.0** | 17.6 | **21.8** |
| MBPP | 28.7 | **19.2** | 23.0 | **32.0** | 20.1 | 24.8 | 30.1 | **21.5** | 25.2 | **34.4** | 22.8 | **27.2** | 35.0 | **24.2** | 28.7 | **38.1** | 25.6 | **30.6** |

Table 7: Performance comparison on the additional datasets not shown in Table 3. Same setup as before, results are reported in terms of precision (P), recall (R), and F1-score, with better scores highlighted in **bold**.

