# OpenReview forum: "Unifying Latent Uncertainty Signals in Large Language Models for Improved Factual Precision"
_ICLR.cc/2026/Conference — ICLR 2026 Conference Withdrawn Submission_

### Official Review · Reviewer_9Cuh · 2025-10-21

**Soundness:** 2
**Presentation:** 2
**Contribution:** 2
**Rating:** 2
**Confidence:** 5

**Summary:**

This paper uses linear probes on hidden layers to predict whether a model is confident on its answers, finding that correctness is reasonably well predicted (at least, in-distribution). However, the paper does not consider the vast literature in uncertainty quantification for LLMs, including a number of probe-based works.

**Strengths:**

* Finds that correctness can be predicted from hidden states via linear probes
* Suggests a reasonable fine-tuning strategy based on explicitly encouraging calibration of hidden layers
* Fine-tuning strategy shows promising results.

**Weaknesses:**

* Lack of discussion of existing literature on probe-based approaches. For example, Kadavath et al (2022) use a form of nonlinear probe; Kapoor et al (2024) and Farquhar et al (2024) use linear probes; Kossen et al (2024) uses semantic entropy based probes (and compare with linear probes). While not a probe-based method, Chen et al (2024) look at uncertainty at different levels of representation. Azaria and Mitchell (2023), Burns et al (2022), and Marks and Tegmark (2024) are also related. Some of these are mentioned in the paper, but only in passing and without commenting on differences.
* Lack of discussion of/comparison with existing work on uncertainty quantification. The authors state “Uncertainty, however, is a concept that LLMs are not generally known to capture”... but there is a lot of work on how well LLMs can capture uncertainty, with several works indicating that LLMs are surprisingly well-calibrated. Outside of the probe-based methods described above, there is a lot of work on extracting uncertainties from models. While I acknowledge that this paper is being presented as (at least partially) an interpretability contribution, I would like to see comparisons with some of the standard UQ methods (eg, semantic entropy, P(True), perplexity...)
* Confusing choices of metric: The authors jump between accuracy, vs precision/recall/F1, and it is hard to compare results.
* Lack of consideration of OOD: While they do look at a probe trained on multiple datasets, there is no consideration of behavior when none of the test distribution is in the train set.
* Lack of experimental details.

Azaria and Mitchell, The Internal State of an LLM Knows When It’s Lying, EMNLP Findings, 2023.
Burns et al, Discovering latent knowledge in language models without supervision. Arxiv, 2022.
Chen et al, INSIDE: LLMs' Internal States Retain the Power of Hallucination Detection. ICLR, 2024.
Farquhar et al, Detecting hallucinations in large language models using semantic entropy. Nature, 2024.
Kadavath et al, Language models (mostly) know what they know. Arxiv, 2022.
Kapoor et al, Large Language Models Must Be Taught to Know What They Don’t Know. NeurIPS 2024.
Kossen et al, Semantic Entropy Probes: Robust and Cheap Hallucination Detection in LLMs. Arxiv, 2024.
Marks and Tegmark, The Geometry of Truth: Emergent Linear Structure in Large Language Model Representations of True/False Datasets. COLM 2024

**Questions:**

* My biggest concern is the lack of positioning within the literature. This would need some significant rewriting of the paper, and discussion of novelty wrt existing work. Given the amount of existing work on probes, I am not sure there is enough novelty for ICLR, but this would be easier to assess with a comprehensive consideration of related work.
* A lot of experimental details are missing, can you elaborate on the following:
    * What is the fine-tuning protocol? #epochs, stopping criteria, etc etc
    * What is the train/val/test split? (Is there a val split? Only a test set is mentioned)
    * How is $\lambda$ selected?
    * How is the training data for the correctness loss obtained — via greedy decoding prior to training? Multiple temp-1 samples? Updated every epoch?


* Can you compare against other UQ-based methods? Eg using AUROC, since many don’t explicitly yield a probability
* Can you make the evaluation metrics consistent between pre- and post-tuned? I would like to see at least accuracy and AUROC in both cases.
* You look at a lot of base models, but only one instruct-tuned model. I don’t think it’s fair to make generalizations about instruct-tuned models being better at correctness prediction based on this.
* Ideally, I’d like to see results on true OOD settings — where the target dataset is not in-distribution. Generally, probes perform poorly in this setting. Does the correctness-augmented fine-tuning approach leads to more or less of a drop in performance outside the training distribution vs basic fine tuning?

---

### Official Review · Reviewer_H3kv · 2025-10-27

**Soundness:** 2
**Presentation:** 1
**Contribution:** 1
**Rating:** 2
**Confidence:** 4

**Summary:**

This paper proposes a method to calibrate the uncertainty of LLMs. The approach consists of two stages. First, a binary linear classifier is trained on the LLM's hidden states to predict whether the model's generated answer is correct. Second, this classifier is frozen, and its binary cross-entropy loss is subsequently used as an auxiliary loss to fine-tune the underlying LLM. The authors posit that this alignment process calibrates the model's uncertainty.

**Strengths:**

1. The paper addresses the critical problem of uncertainty in LLMs, which is a key barrier to their reliable deployment in high-risk, safety-critical applications.

2. The proposed method appears straightforward to implement and computationally efficient. It requires only training a binary classifier on the LLM's hidden states and then using the binary cross-entropy as an auxiliary loss for fine-tuning. This approach avoids complex architectural modifications and significant computational overhead.

**Weaknesses:**

## Weaknesses

1. The core technical approach trains a binary classifier to predict *correctness*, but this is used as a proxy for *uncertainty*. These two concepts are not interchangeable. An incorrect answer can stem from both overconfidence (low uncertainty) and underconfidence (high uncertainty), just as a correct answer does not necessarily imply high certainty. Therefore, the vector $\mathbf{u}_i$ derived in Equations (1) and (2) likely represents a direction of "correctness" rather than a reliable measure of uncertainty.

2. Given that the auxiliary loss directly optimizes the classifier's objective (BCE), it is unsurprising that the "aligned" model improves the classifier's F1 score. (I assume Tables 2 and 3 report the performance of this *classifier* rather than the LLM itself, please clarify if this is not the case). However, the paper fails to establish three critical points:
    * 2.1 It is unclear how this classifier's predicted score translates into a usable uncertainty metric for the LLM's generation.

    * 2.2 It is unclear whether this score is well-calibrated (e.g., evaluated using metrics like ECE or Brier Score), which is essential for any practical uncertainty measure.

    * 2.3 It is unclear whether the alignment fine-tuning degrades the LLM's task accuracy on the original QA benchmarks. This analysis is missing.

3. The ablation study is limited. Crucially, it lacks an analysis of the sensitivity to the hyperparameter $\lambda$, which balances the auxiliary loss and the primary model loss.

4. The paper's quality of writing and presentation requires significant revision. The Introduction dedicates excessive space to basic concepts (e.g., hallucination) instead of motivating the specific challenge the paper addresses. The Introduction also fails to summarize the main quantitative results, leaving the reader without a clear understanding of the method's performance. Furthermore, the acronym "LLM" is defined twice in the Introduction. In Figure 2, the color scheme and small font size make the results very difficult to read.

5. The paper's premise of using a linear probe on hidden states seems to overlook findings from related work. For instance, Wang et al. [1] have already demonstrated the limitations of using universal direction vectors for related tasks like hallucination detection. Their analysis (e.g., Figure 2 in [1]) appears more rigorous than the analysis in this work.

6. The related work section is incomplete. It does not discuss a relevant and important line of research on Bayesian methods for LLM uncertainty and calibration, such as [2, 3, 4].

---

[1] Wang, Hanyu, et al. "Truthflow: Truthful llm generation via representation flow correction." ICML 2025.

[2] Yang, Adam X., et al. "Bayesian Low-rank Adaptation for Large Language Models." ICLR 2024.

[3] Wang, Yibin, et al. "Blob: Bayesian low-rank adaptation by backpropagation for large language models." NeurIPS 2024.

[4] Li, Yawei, et al. "Calibrating LLMs with Information-Theoretic Evidential Deep Learning." ICLR 2025.

**Questions:**

Please see the Weaknesses.

---

### Official Review · Reviewer_w4v9 · 2025-10-31

**Soundness:** 3
**Presentation:** 3
**Contribution:** 1
**Rating:** 2
**Confidence:** 4

**Summary:**

This study employs probing techniques to investigate the internal uncertainty of the model. Concurrently, a training method is designed, which involves aligning the defined uncertainty with the ground-truth uncertainty to predict the "accuracy rate of correct model responses".

**Strengths:**

- This manuscript conducts an in-depth exploration of "uncertainty"—a concept of significant importance in the field.​
- The paper proposes a method for predicting the model's accuracy rate, and this method demonstrates certain effectiveness in practical applications.

**Weaknesses:**

* The authors may have omitted some key citations, such as

  * Ji, Ziwei, et al. "Calibrating Verbal Uncertainty as a Linear Feature to Reduce Hallucinations." arXiv preprint arXiv:2503.14477 (2025).

  * Zhang, Caiqi, et al. "Reinforcement Learning for Better Verbalized Confidence in Long-Form Generation." arXiv preprint arXiv:2505.23912 (2025).


* The application value proposed by the authors—predicting the model's own accuracy rate—does not seem to have received adequate evaluation. In my opinion, it would be more appropriate to adopt methods such as ECE and EUROC for this purpose.

**Questions:**

Have the authors considered how to explicitly communicate uncertainty to users in a more user-friendly manner? For instance, by adjusting the tone of the model's output?​

I remain of the opinion that obtaining uncertainty through external tools is not an optimal approach for end-user-oriented scenarios.

---

### Official Review · Reviewer_1Y7a · 2025-11-10

**Soundness:** 1
**Presentation:** 1
**Contribution:** 1
**Rating:** 2
**Confidence:** 3

**Summary:**

1. Authors explore and analyze the performance characteristics of probe-based UQ methods for NLG tasks.
2. Authors propose a finetuning methodology with the intention of improving LLM's ability to abstain when it's uncertain.

**Strengths:**

S1. I appreciate the empirical evaluation on multiple datasets and distinct families of LLMs.

**Weaknesses:**

W1. Authors do not engage with prior work closely related to the subject. The most important work that comes to my mind that the authors should engage with is https://arxiv.org/abs/2410.02707 (ICLR 2025), which similar observations regarding limited transfer of uncertainty-probes across problem distributions (similar observations are made by https://arxiv.org/abs/2506.08572 but this can be conidered concurrent work), and takeaways from Fig. 5 while at the same time additionally explores the subject of the token position from which the embedding is obtained.



W2. Incorrect conclusions about "a generalized axis of uncertainty" in Sec 2, L247-252.
For the dimensionalities of the problem studied in Fig 3, taking 17 fixed random unit vectors (equal to number of datasets in Fig 3), and finding a unit vector maximizing the average cos-similarity (in closed form), we find that the average cos-similarity achieveable is ~0.24. See code at the bottom.
This implies that the results in Fig 3 can occur for completely random vectors and don't imply existance of "a generalized axis of uncertainty".
I might have misunderstood something or made a mistake in the code, so feel free to investigate, and convince me otherwise.

```
import torch
import matplotlib.pyplot as plt

N = 1000
n_datasets = 17
embedding_dims = 4096

fixed_unit_vectors = torch.randn(N, n_datasets, embedding_dims)
fixed_unit_vectors /= torch.linalg.norm(fixed_unit_vectors, dim=-1, keepdim=True)

vsum = fixed_unit_vectors.sum(dim=1, keepdim=True)
optimal_vec = vsum / torch.linalg.norm(vsum, dim=-1, keepdim=True)  # [N, 1, embedding_dims]

cos_sims = torch.bmm(fixed_unit_vectors, optimal_vec.transpose(-1, -2))  # [N, n_datasets, 1]

plt.hist(cos_sims.reshape(-1).numpy())
print(cos_sims.mean())  # ~0.24
```

W3. Evaluation methodology.
W3a. Table 1 & Figure 2: accuracy alone cannot be interpreted in any reasonable way because the baseline accuracy (for a random predictor) is not provided.
Please report your results using AUROC (which is a fixed-baseline metric), which is the best metric for this setting and widely adapted in related literature.

W4. The impact of finetuning proposed is small and does not seem to generalize well. Additionally it is not clear what other performance impacts this finetuning procedure has (e.g. perplexity on Wikipedia).

**Questions:**

Remarks

R1. Make the diagonal of Fig 3 a different color, and extend the colormap used for the heatmap to capture the entire range of values plotted. Currently, using the same color for the entire range 0.2-1.0 is extremely misleading.

Questions

Q1. L76-78: "the model is often also encouraged to refrain from answering questions when the specific answer is not known to it." - could the authors provide some reference or source for this, please?

Q2. Do I understand correctly that Linear Uncertainty Alignment updates all of the weights of the LLM? i.e. no weights are frozen? (this is similar to P(IK) method of https://arxiv.org/abs/2207.05221)

Q3. L363: What do the authors mean by "calibrated use of abstentions"? How do the authors evaluate whether the "use of abstensions" is calibrated? Same for L407: "the mode becomes more calibrated", and L318-319. I'm asking about it because "calibration" has pretty specific technical meaning and I want to understand how the authors understand it in this setting (because I don't understand how it applies here).

Q4. It is not clear to me - what exactly does "Tuned" mean in Tables 2 & 3? Is it a) the original model, or b) the variant 1 in "Tuning Variants"? If the latter, I would like to request the authors also report the performance of the original model without any finetuning by the authors, please.

Q5. I am not fully certain I understand what Precision and Recall authors report in Tables 2-4 - could the authors very explicitly state how those are defined, please?

---

### Note · Authors · 2025-12-03

I have read and agree with the venue's withdrawal policy on behalf of myself and my co-authors.